# New-Onset Type 1 Diabetes in a Child with Joubert Syndrome: A Rare Endocrine Complication

**DOI:** 10.3390/reports8020057

**Published:** 2025-04-27

**Authors:** Yutaka Furuta, Erica T. Nelson, Rory J. Tinker, Angela R. Grochowsky

**Affiliations:** 1Department of Pediatrics, Division of Medical Genetics and Genomic Medicine, Vanderbilt University Medical Center, Nashville, TN 37232, USA; 2Department of Pediatrics, Division of Genetic and Genomic Medicine, Nationwide Children’s Hospital, Columbus, OH 43205, USA

**Keywords:** ciliopathy, Joubert syndrome, type 1 diabetes

## Abstract

**Background and Clinical Significance:** Joubert syndrome (OMIM #213300) is a rare predominantly autosomal recessive inherited condition characterized by the classic cerebellar vermis hypoplasia and brainstem anomalies (also known as the “molar tooth sign”), hypotonia, and developmental delays. Joubert syndrome is a ciliopathy that affects multiple systems including the central nervous system, eyes, kidneys, liver, respiratory, musculoskeletal system, cardiovascular system, and endocrine system. Endocrine abnormalities are not uncommon in Joubert syndrome, such as growth hormone deficiency, thyroid hormone deficiency, central diabetes insipidus, hypopituitarism, micropenis, and obesity. However, a new-onset type 1 diabetes in childhood is not common in Joubert syndrome. **Case Presentation:** Herein, we report a case of a 7-year-old male with a history of Joubert syndrome presenting with polydipsia, polyuria, weight loss, and hyperglycemia who was diagnosed with type 1 diabetes. **Conclusions:** While diabetes has been reported as a rare complication in Joubert syndrome, this is the first case report of Joubert syndrome to accentuate new-onset type 1 diabetes as an endocrine complication.

## 1. Introduction

Joubert syndrome (JS; OMIM #213300) is a predominantly autosomal recessive inherited condition characterized by a distinctive cerebellar and brainstem defect on a magnetic resonance imaging (MRI) known as the “molar tooth sign.” JS is a ciliopathy that presents with a broad spectrum of phenotypes due to primary cilium dysfunction. Other characteristic features include hypotonia, developmental delay, ataxia, oculomotor apraxia, retinal dystrophy, renal cysts, hepatic fibrosis, abnormal breathing pattern with episodic tachypnea and/or apnea, and polydactyly [1]. The inheritance pattern of JS is predominantly autosomal recessive while a few cases of an X-linked pattern have been reported [2]. To date, pathogenic variants in more than 35 genes have been reported in JS, commonly including *INPP5E*, *TMEM216*, *AHI1*, *NPHP1*, *CEP290 (NPHP6)*, *TMEM67 (MKS3)*, *RPGRIP1L*, *ARL13B*, *CC2D2A*, *OFD1*, *TTC21B*, *KIF7*, *TCTN1*, *TCTN2*, *TMEM237*, *CEP41*, *TMEM138*, *C5orf42*, *TCTN3*, *ZNF423*, *TMEM231*, *CSPP1*, and *PDE6D* [3].

One of the known complications of JS is endocrine abnormalities. These affect an estimated 4–11% of patents [2,4]. These abnormalities include but are not limited to pituitary hormone dysfunction (growth hormone deficiency, thyroid hormone deficiency, central diabetes insipidus, and panhypopituitarism), micropenis, premature puberty, ovarian insufficiency, and obesity. While a review by Bachmann-Gagescu et al. reports diabetes as an endocrine complication in JS [5], diabetes is not recognized in most published review articles. To our knowledge, there is only one previous case report of a patient with a history of JS who presented with new-onset type 1 diabetes and hyperglycemic hyperosmolar state [6]. However, no correlation between JS and diabetes was described in this report. We report here a case of JS with new-onset type 1 diabetes and discuss ciliopathies that can cause diabetes.

## 2. Case Presentation

A 7-year-old male with a history of Joubert syndrome (JS), global developmental delay, hypotonia, dysarthria, ataxia, oculomotor apraxia, and autism, presented to the emergency department with polydipsia, polyuria, weight loss, and hyperglycemia. He was in his usual state of health until three weeks prior to the visit when he had a viral upper respiratory infection with fever, cough, and nasal congestion. These symptoms had resolved, but he developed polydipsia, polyuria, and fatigue a week prior to the visit. His parents reported he also had weight loss. He had urinary incontinence with a syncopal episode at school, which prompted his presentation to the emergency department.

In the emergency department, he was alert but ill-appearing. He was not obese (weight: 20.5 kg, 15.8 percentile, Z = −1.00). His vital signs were notable for mild tachycardia. Laboratory studies showed hyperglycemia of 396 mg/dL in the setting of uncompensated metabolic acidosis (pH 7.33, pCO2 35 mmHg, bicarbonate 18.5 mEq/L), with an anion gap of 18 mmol/L. His urinalysis was positive for glucose (>1000 mg/dL) and ketones (>150 mg/dL). These findings did not meet the criteria for diabetic ketoacidosis per the hospital’s clinical practice guidelines. He was admitted for further management of hyperglycemia.

On review of his medical history, he was born at 42 weeks via cesarean section due to failed postdates induction. At the age of 32 months, he was referred to a genetics clinic for concern of global developmental delay, hypotonia, dysarthria, ataxia, oculomotor apraxia, and the characteristic cerebellar vermis hypoplasia with “molar tooth sign” on brain MRI (Table 1). He was then diagnosed with JS via a ciliopathies panel revealing two pathogenic variants, c.424G>A (*p*.Glu142Lys) and c.178T>G (*p*.Leu595*), in *CPLANE1*. The panel also revealed a pathogenic variant in *DHAH5* as well as a variant of uncertain significance in *CEP290* and *DNAH11*. On review of the family history, his older sister was diagnosed with JS after his diagnosis. His maternal grandparents have type 2 diabetes while no one has type 1 diabetes.

He was admitted to the hospital for initiation of insulin (long-acting insulin and insulin lispro) and new-onset diabetes education. His new-onset type 1 diabetes testing returned and revealed positive glutamic acid decarboxylase antibody (121.5 IU/mL; reference range 0–0.5 IU/mL), reduced C-peptide level (0.6 ng/mL; reference 0.7–5.2 ng/mL), and an elevated hemoglobin A1c of 7.1%, which confirmed the diagnosis of type 1 diabetes. He was discharged with improvement in his hyperglycemia following the initiation of an insulin regimen.

## 3. Discussion

Here we describe a 7-year-old male with JS who was found to have new-onset type 1 diabetes. JS (OMIM #213300) is a ciliopathy condition that involves multiple systems, particularly including the central nervous system, the eyes, and the respiratory. Interestingly, endocrinologic complications are absent from the phenotypic description in OMIM #213300 (Table 1). A subset of individuals with JS also has musculoskeletal, renal, hepatic, cardiovascular, and endocrine complications [5]. Endocrine abnormalities have been described in JS, including central diabetes insipidus, premature puberty, hypothyroidism, growth hormone deficiency, panhypopituitarism, micropenis, and obesity [5]. Morelli et al. reported that 7 of 59 (11%) children with JS presented with endocrine conditions (mainly short stature) [4]. Bachmann-Gagescu et al. reported the prevalence of endocrine complications as 4–6%, which occur from infancy to adulthood [5]. They specifically reported type 1 diabetes, hypothyroidism, and polycystic ovarian syndrome in 1–2% of JS cases. However, this may be related to the prevalence in the general population. Obesity is thought to be increased in JS patients as it is in other ciliopathies like Bardet–Biedl syndrome. This association is strengthened by pathogenic variants in *INPP5E* accounting for both JS and MORM syndrome (mental retardation, obesity, retinal dystrophy, and micropenis) [7,8].

To date, only one case report of JS with type 1 diabetes has been previously published, in which a 7-year-old male with JS was diagnosed with new-onset type 1 diabetes and hyperglycemic hyperosmolar state (HHS) [6]. However, this report focused on understanding the pathophysiology and management of HHS, not on JS. Therefore, our case is the first report to highlight type 1 diabetes as a possible endocrine complication of JS. More specifically, *CPLANE1* pathogenic variants are detected in 8–14% of JS cases, including our patient, but no correlation with diabetes has been reported [2,9,10].

The pancreas consists of exocrine and endocrine compartments. The endocrine pancreatic beta cells play a central role in maintaining glucose homeostasis. The autoimmune destruction of these cells leads to a deficiency of insulin which is the mechanism behind type 1 diabetes. On the other hand, type 2 diabetes is due to insulin resistance and the inability of beta cells to compensate for the resistance [11]. In relation to ciliopathies, like JS, primary cilia are present in the beta, alpha, and delta cells. They play crucial roles in regulating key various signaling pathways, including fibroblast growth factors (FGF), Sonic Hedgehog (Shh), Wnt, and Notch, which are essential for pancreatic development and function [12]. Even so, the mechanism of diabetes associated with ciliopathies is not fully understood; it is suggested ciliary dysfunction is implicated in insulin signaling and secretion in the beta cells, resulting in type 2 diabetes [13]. However, this does not explain the pathophysiology of type 1 diabetes.

Although the mechanism of diabetes in JS is still unclear, it is important to recognize other ciliopathies in which diabetes is a common feature. Ciliopathies share clinical features such as mental retardation, cystic kidney disease, retinal defects, polydactyly, obesity, and diabetes [14]. Some ciliopathies present with obesity and diabetes such as Alström syndrome (ALMS; OMIM #203800) and Bardet–Biedl syndrome (BBS; OMIM #209900).

ALMS and BBS are autosomal recessively inherited ciliopathies with common characteristics of obesity and diabetes. While both exhibit significant obesity, they demonstrate marked distinct rates of childhood diabetes, with ALMS having a rate of 75% and BBS ranging from 2–6% [12]. ALMS presents with early-onset diabetes, whereas BBS is characterized by delayed-onset type 2 diabetes [15,16]. The observed difference in the incidence of diabetes in ALMS and BBS, despite similar levels of obesity, suggests the possibility that diabetes occurrence may be independent of obesity [12]. Lodh et al. suggest that this discrepancy implies that distinctions in the maintenance of beta cell population or function could rather contribute to the varied impact of T2DM in these symptoms, with T2DM affecting a smaller proportion of patients and with BBS and having a later age of onset compared to patients with ALMS [12]. Several reports have described patients with BBS who were diagnosed with type 1 diabetes [17,18,19], despite the distinct pathogenesis of type 1 and type 2 diabetes. Further research is needed to clarify the exact mechanism underlying the development of type 1 diabetes in ciliopathies, including JS.

## 4. Conclusions

In summary, this is the first case report of Joubert syndrome associated with new-onset type 1 diabetes, a rare endocrine complication. Type 1 diabetes is not recognized in most published review articles on Joubert syndrome. It is important for clinicians to be aware that type 1 diabetes can occur in patients with Joubert syndrome.

## Figures and Tables

**Table 1 reports-08-00057-t001:** Comparison of clinical features of patients with Joubert syndrome reported in OMIM 213,300 and our patient.

	OMIM Reported Features	Our Patient
Eyes	Abnormal jerky eye movement, impaired smooth pursuit, impaired saccades, oculomotor apraxia, coloboma, retinal dystrophy	Oculomotor apraxia
Respiratory	Breathing dysregulation, apnea	No
Abdomen	Hepatic fibrosis	No
Endocrinology	Not mentioned	Type 1 diabetes
Genitourinary	Renal cysts	No
Skeletal	Missing digital phalanges, polydactyly	No
Neurologic	Molar tooth sign, hypotonia, ataxia	Molar tooth sign, hypotonia, dysarthria, ataxia
Development	Delayed psychomotor development, mental retardation, hyperactivity, aggressiveness, self-mutilation	Global developmental delay, autism

## Data Availability

The original contributions presented in this study are included in the article. Further inquiries can be directed to the corresponding author.

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
