# Peer review of "New-Onset Type 1 Diabetes in a Child with Joubert Syndrome: A Rare Endocrine Complication"

_reports, 2025, doi:10.3390/reports8020057_

Round 1
Reviewer 1 Report
Comments and Suggestions for Authors
I would like to congratulate the authors on this valuable work, which represents a significant contribution to the medical literature by exploring the relationship between Joubert syndrome and the onset of type 1 diabetes in childhood. The detailed description of the clinical case is particularly relevant, given that this association is rare and poorly documented. The inclusion of this case report expands the understanding of possible endocrine complications in patients with this syndrome, allowing the medical community to consider type 1 diabetes as an additional clinical manifestation in these patients.
Since type 1 diabetes is an autoimmune disease with multiple triggering factors, it is pertinent to evaluate the possibility that a prior viral infection may have acted as a predisposing factor in this case. The literature has reported the role of various viruses, such as enteroviruses (particularly Coxsackie B4), cytomegalovirus, parvovirus B19, and Epstein-Barr virus, in triggering autoimmune responses directed against pancreatic beta cells in genetically susceptible individuals. In this regard, it would be of interest to delve into the patient’s medical history to determine whether they had any of these viral infections in the weeks preceding the diagnosis of type 1 diabetes, as the manuscript mentions that the patient suffered from a viral infectious episode before the onset of hyperglycemic symptoms. Additionally, it would be relevant to consider other infectious agents that may be implicated in immune system disruption, particularly in individuals with syndromic diseases, who may have a higher risk of autoimmune or metabolic dysfunction due to alterations in immune regulation. It would be advisable to include this information again in the manuscript to enhance the understanding of the possible pathogenic link.
Furthermore, evaluation by a pediatric endocrinology specialist and specific studies for the detection of autoimmune markers are recommended to better characterize the underlying mechanism of type 1 diabetes in this clinical context. Since the pathogenesis of this disease is multifactorial, involving genetic, immunological, and environmental factors, a comprehensive approach would be essential to better characterize this association and understand its impact on the patient’s clinical course.
Regarding future research directions, I suggest expanding the case series to assess whether more patients with Joubert syndrome develop type 1 diabetes. Additionally, conducting immunogenetic studies could contribute to the identification of biomarkers associated with this endocrine complication. Finally, investigations into prior viral infections in patients with Joubert syndrome and type 1 diabetes could help establish potential epidemiological and pathogenic correlations, which may contribute to the development of preventive strategies in at-risk populations.
Once again, I congratulate the authors on this important contribution, which will undoubtedly enhance the understanding of the endocrine spectrum of Joubert syndrome and its possible relationship with autoimmunity.
Author Response
Comment 1: Since type 1 diabetes is an autoimmune disease with multiple triggering factors, it is pertinent to evaluate the possibility that a prior viral infection may have acted as a predisposing factor in this case. The literature has reported the role of various viruses, such as enteroviruses (particularly Coxsackie B4), cytomegalovirus, parvovirus B19, and Epstein-Barr virus, in triggering autoimmune responses directed against pancreatic beta cells in genetically susceptible individuals. In this regard, it would be of interest to delve into the patient’s medical history to determine whether they had any of these viral infections in the weeks preceding the diagnosis of type 1 diabetes, as the manuscript mentions that the patient suffered from a viral infectious episode before the onset of hyperglycemic symptoms. Additionally, it would be relevant to consider other infectious agents that may be implicated in immune system disruption, particularly in individuals with syndromic diseases, who may have a higher risk of autoimmune or metabolic dysfunction due to alterations in immune regulation. It would be advisable to include this information again in the manuscript to enhance the understanding of the possible pathogenic link.
Response 1: We thank the comment. We strongly agree that it is important to evaluate for an infectious etiology. It is unclear whether the patient underwent infectious testing, such as a respiratory viral panel, during his viral symptoms. We did not obtain a respiratory viral panel in our hospital because his viral symptoms had already resolved.
Comment 2: Furthermore, evaluation by a pediatric endocrinology specialist and specific studies for the detection of autoimmune markers are recommended to better characterize the underlying mechanism of type 1 diabetes in this clinical context. Since the pathogenesis of this disease is multifactorial, involving genetic, immunological, and environmental factors, a comprehensive approach would be essential to better characterize this association and understand its impact on the patient’s clinical course.
Response 2: We thank the comment. We conducted autoimmune and endocrinologic evaluations to establish a diagnosis of type 1 diabetes. Please see the following updated text.
Updated text: His new-onset type 1 diabetes testing returned and revealed positive glutamic acid decarboxylase antibody (121.5 IU/mL; reference range 0-0.5 IU/mL), reduced C-peptide level (0.6 ng/mL; reference 0.7-5.2 ng/mL), and elevated hemoglobin A1c of 7.1%, which confirmed the diagnosis of type 1 diabetes.
Reviewer 2 Report
Comments and Suggestions for Authors
I read with great interest your case report "New-Onset Type 1 Diabetes in a Child with Joubert Syndrome: A Rare Endocrine Complication". The case is very interesting.
Did you obtain ethical committee approval, if so could you please mention its number.
Was informed written consent taken from the legal guardians, if so please mention this.
How did you prove it's type 1 diabetes, were antibodies done, fasting c peptide assessed?
How was the ABG, it would be nice to add this.
Hyperglycaemic hyperosmolar coma is quite rare it T1DM, DKA is more common due to absolute insulin deficiency. How do you explain your case?
Was the patient controlled on insulin?
What type of insulin did he receive?
Was CGM done?
Author Response
Comment 1: Did you obtain ethical committee approval, if so could you please mention its number.
Response 1: We thank the comment. The ethical committee approval was not required in this case.
Comment 2: Was informed written consent taken from the legal guardians, if so please mention this.
Response 2: We thank the comment. Verbal informed consent for publication was obtained from the patient’s parent. We made an attempt to obtain written informed consent, but were unable to do so due to the patient’s distance from our facility.
Comment 3: How did you prove it's type 1 diabetes, were antibodies done, fasting c peptide assessed?
Response 3: We thank the comment. Please see the following updated text.
Updated text: His new-onset type 1 diabetes testing returned and revealed positive glutamic acid decarboxylase antibody (121.5 IU/mL; reference range 0-0.5 IU/mL), reduced C-peptide level (0.6 ng/mL; reference 0.7-5.2 ng/mL), and elevated hemoglobin A1c of 7.1%, which confirmed the diagnosis of type 1 diabetes.
Comment 4: How was the ABG, it would be nice to add this.
Response 4: We thank the comment. VBG was obtained in the emergency room instead of ABG. Please see the following updated text.
Updated text: Laboratory studies showed hyperglycemia of 396 mg/dL in the setting of uncompensated metabolic acidosis (pH 7.33, pCO2 35 mmHg, bicarbonate 18.5 mEq/L), with an anion gap of 18 mmol/L.
Comment 5: Hyperglycaemic hyperosmolar coma is quite rare it T1DM, DKA is more common due to absolute insulin deficiency. How do you explain your case?
Response 5: We thank the comment. We agree the reviewer’s insight. In this case, the patient has been alert since the initial presentation.
Comment 6: Was the patient controlled on insulin?
Response 6: We thank the comment. The patient’s hyperglycemia was controlled well with insulin. Please see the following updated text.
Updated text: He was discharged with improvement in his hyperglycemia following the initiation of an insulin regimen.
Comment 7: What type of insulin did he receive?
Response 7: We thank the comment. Please see the following updated text.
Updated text: He was admitted to the hospital for initiation of insulin (long-acting insulin and insulin lispro) and new-onset diabetes education.
Comment 8: Was CGM done?
Response 8: We thank the comment. We only performed scheduled glucose checks and did not conduct continuous glucose monitoring (CGM) in this case.
Reviewer 3 Report
Comments and Suggestions for Authors
Thank you for the opportunity to review the case presentation of Furuta et al which presents the rare association of Joubert syndrome with type 1 diabetes. As the authors state, this is the 2nd case of this kind reported in the literature and the aim of such case presentations is to draw attention of the medical community on uncommon associations of different conditions. Overall, the manuscript is very well written and interesting. The Introduction provides sufficient details on what is currently known about the Joubert syndrome, which is useful for the healthcare providers which are not specialists in the subject. While the Case presentation tries to provide details on the case and the Discussion section brings data on what is currently known on the association of diabetes with the Joubert syndrome, ciliopathies and diabetes there are some aspects which could be improved.
- For the case presentation could the authors provide additional details on the laboratory results at presentation? Now the authors state briefly glycemia, and the fact that metabolic acidosis and ketonuria were present, while the criteria for diabetic ketoacidosis were not met. Which is odd given the ISPAD criteria for DKA. It would be particularly interesting to see the pH, HCO3 levels and ketonemia/ketonuria levels which would support the authors’ statement.
- In the Discussion section, the authors discuss mainly the association of Joubert syndrome and other ciliopathies with type 2 diabetes. The pathogenetic mechanisms of diabetes type 2 and type 1 are different and the current discussion does not support the association observed. However, there are reports in the literature that support the association of ciliopathies with autoimmune diseases. The authors should discuss this association which may explain the coexistence of the Joubert syndrome with type 1 diabetes in the case presented.
Author Response
Comment 1: For the case presentation could the authors provide additional details on the laboratory results at presentation? Now the authors state briefly glycemia, and the fact that metabolic acidosis and ketonuria were present, while the criteria for diabetic ketoacidosis were not met. Which is odd given the ISPAD criteria for DKA. It would be particularly interesting to see the pH, HCO3 levels and ketonemia/ketonuria levels which would support the authors’ statement.
Response 1: We thank the comment. We added the laboratory values. Please see the following updated text.
Updated text: Laboratory studies showed hyperglycemia of 396 mg/dL in the setting of uncompensated metabolic acidosis (pH 7.33, pCO2 35 mmHg, bicarbonate 18.5 mEq/L), with an anion gap of 18 mmol/L. His urinalysis was positive for glucose (>1,000 mg/dL) and ketones (>150 mg/dL).
Comment 2: In the Discussion section, the authors discuss mainly the association of Joubert syndrome and other ciliopathies with type 2 diabetes. The pathogenetic mechanisms of diabetes type 2 and type 1 are different and the current discussion does not support the association observed. However, there are reports in the literature that support the association of ciliopathies with autoimmune diseases. The authors should discuss this association which may explain the coexistence of the Joubert syndrome with type 1 diabetes in the case presented.
Response 2: We thank the comment. As the reviewer points out, type 2 diabetes is much rarer in ciliopathies. One possible reason is that many ciliopathies present with obesity, while the exact mechanism of diabetes associated with ciliopathies remains not fully understood. At present, it is unclear why type 1 diabetes can occur in ciliopathies, despite the differing pathophysiology, as the reviewer notes. We have added additional references to strengthen the discussion.
Update texts: Several reports have described patients with BBS who were diagnosed with type 1 diabetes [17-19], despite the distinct pathogenesis of type 1 and type 2 diabetes. Further research is needed to clarify the exact mechanism underlying the development of type 1 diabetes in ciliopathies, including JS.
[17] Halac U, Herzog D. Bardet-Biedl Syndrome, Crohn Disease, Primary Sclerosing Cholangitis, and Autoantibody Positive Thyroiditis: A Case Report and A Review of a Cohort of BBS Patients. Case Rep Med. 2012;2012:209827. doi: 10.1155/2012/209827. Epub 2012 Aug 15. PMID: 22927860; PMCID: PMC3426198.
[18] Elawad OAMA, Dafallah MA, Ahmed MMM, Albashir AAD, Abdalla SMA, Yousif HHM, Daw Elbait AAE, Mohammed ME, Ali HIH, Ahmed MMM, Mohammed NFN, Osman FHM, Mohammed MAY, Abu Shama EAE. Bardet-Biedl syndrome: a case series. J Med Case Rep. 2022 Apr 29;16(1):169. doi: 10.1186/s13256-022-03396-6. PMID: 35484558; PMCID: PMC9052695.
[19] Aleman TS, O'Neil EC, O'Connor K, Jiang YY, Aleman IA, Bennett J, Morgan JIW, Toussaint BW. Bardet-Biedl syndrome-7 (BBS7) shows treatment potential and a cone-rod dystrophy phenotype that recapitulates the non-human primate model. Ophthalmic Genet. 2021 Jun;42(3):252-265. doi: 10.1080/13816810.2021.1888132. Epub 2021 Mar 17. PMID: 33729075; PMCID: PMC8743897.
Round 2
Reviewer 2 Report
Comments and Suggestions for Authors
All comments were addressed properly. No further comments required.
Reviewer 3 Report
Comments and Suggestions for Authors
No additional comments.